# Comparison of the Impact of Pharmacogenetic Variability on the PK of Slow Release and Immediate Release Tacrolimus Formulations

**DOI:** 10.3390/genes11101205

**Published:** 2020-10-15

**Authors:** Teun van Gelder, Oumaima Etsouli, Dirk Jan Moes, Jesse J. Swen

**Affiliations:** Department of Clinical Pharmacy & Toxicology, Leiden University Medical Center, Albinusdreef 2, 2333 Leiden, The Netherlands; o.etsouli@lumc.nl (O.E.); d.j.a.r.moes@lumc.nl (D.J.M.); j.j.swen@lumc.nl (J.J.S.)

**Keywords:** tacrolimus, pharmacokinetics, CYP3A, immediate release, modified release, switching, transplantation, therapeutic drug monitoring, pharmacogenetics

## Abstract

Tacrolimus-modified release formulations allow for once-daily dosing, and adherence is better compared to the twice-daily immediate release formulation. When patients are switched from one formulation to another, variable changes in drug concentrations are observed. Current data suggest that the changes in drug exposure are larger in patients who express the CYP3A5 enzyme (*CYP3A5 *1/*3 or *1/*1*) compared to nonexpressers (*CYP3A5*3/*3*). Possibly, these differences are due to the fact that in the upper region of the small intestine CYP3A activity is higher, and that this expression of CYP3A decreases towards the more distal parts of the gut. Modified release formulations may therefore be subject to a less presystemic metabolism. However, the full implications of pharmacogenetic variants affecting the expression and function of drug transporters in the gut wall and of enzymes involved in phase I and phase II metabolism on the different formulations are incompletely understood, and additional studies are required. Conclusions: In all patients in whom the formulation of tacrolimus is changed, drug levels need to be checked to avoid clinically relevant under- or overexposure. In patients with the *CYP3A5* expresser genotype, this recommendation is even more important, as changes in drug exposure can be expected.

## 1. Introduction

Tacrolimus is one of the most frequently used immunosuppressive drugs in the prevention of rejection after a solid organ transplantation. For a review on the clinical development of tacrolimus and the potential advantages of the subsequently developed alternative formulations, we refer to the review paper of Tremblay and Alloway [1]. Tacrolimus is metabolized by demethylation and hydroxylation by the CYP3A4 and CYP3A5 enzymes, in both the gut wall and in the liver. It has been convincingly shown that CYP3A5 expressers (about 15% of Caucasians, but close to half of Asian and African Americans) require higher tacrolimus doses to reach target concentrations compared to patients who depend on CYP3A4 only for their metabolism [2]. The Clinical Pharmacogenetics Implementation Consortium (CPIC) and Dutch Pharmacogenetics Working group have provided dosing recommendations for tacrolimus based on the *CYP3A5* genotype [3,4]. Furthermore, besides covariates such as age, gender and body weight, the *CYP3A5* genotype has been included in dosing algorithms to select the best tacrolimus starting dose for each individual patient following kidney transplantation [5].

The literature that forms the basis for these dose recommendations is largely based on pharmacogenetic studies that have been performed in patients treated with the immediate release tacrolimus formulation. This formulation needs to be taken twice daily and is known under the trade name Prograf^®^ (referred to, in this manuscript, as tacrolimus immediate release). However, in more recent years, in order to improve adherence, modified release formulations of tacrolimus have been registered, allowing for dosing once a day [6,7]. The first on the market was a prolonged release formulation marketed under the trade name Advagraf^®^ (in the United States, Astagraf XL^®^), a capsule containing intermediate-sustained-release granules consisting of a mix of tacrolimus with ethylcellulose, hypromellose and lactose. Several years later, this was followed by a second modified release formulation using the so-called “solid solution” (or MeltDose) delivery technology (Envarsus^®^) [8,9]. The objective of this review is to discuss the effect of the CYP3A5 genotype on changes in exposure to tacrolimus when patients are switched from one formulation to another (once daily vs. twice daily). Before discussing the studies that investigated the impact of pharmacogenetic variability on the pharmacokinetics of the different tacrolimus formulations, we explain why differences might be expected between immediate release and modified release formulations.

### 1.1. Intestinal Distribution of CYP3A Enzymes and Effects on Bioavailability

The bioavailability of tacrolimus is influenced by the presence of CYP3A enzymes in both the intestinal wall and in the liver [10]. For systemic clearance, the activity of intestinal CYP3A is negligible and largely depends on CYP3A activity in the liver. However, as part of the first pass effect, a substantial proportion of the drug is metabolized in the intestinal wall after oral administration. The activity of CYP3A is not equal along the entire length of the gut wall. In the upper region of the small intestine, CYP3A activity is higher, and it decreases towards the more distal part of the small intestine and the colon [11]. In a study using mucosa isolated from duodenal, jejunal and ileal sections of 20 human donor intestines, it was shown that the CYP3A content and catalytic activity was almost two-fold higher in the duodenum than in the ileum (31 vs. 17 pmol/mg of protein) [12]. For both CYP3A4 and CYP3A5, a higher expression has been reported in the proximal parts of the small bowel (jejunum) than in the more distal parts (ileum) [13]. Therefore, the modified release formulations may release most of the tacrolimus into parts of the gut with a lower abundance of CYP3A, potentially bypassing part of the CYP3A-mediated first pass metabolism. As a result, bioavailability may be higher. For Envarsus^®^, a higher bioavailability has indeed been demonstrated. Rostaing et al. [14] showed, in patients, that two years after kidney transplantation the mean total daily dose for Envarsus^®^ was 24% lower than for the tacrolimus immediate release (*p* < 0.001), while the trough concentrations were similar (means of 5.5 and 5.8 ug/L, respectively). For the Advagraf^®^ formulation, a higher bioavailability has not been demonstrated, but Advagraf^®^ is also not bioequivalent to tacrolimus immediate release [15]. In fact, patients may require a small daily dosage increase if converted from tacrolimus immediate release to Advagraf^®^, while a daily dosage reduction appears necessary for conversion from tacrolimus immediate release to Envarsus^®^. In a specifically designed head-to-head two-sequence, three-period crossover pharmacokinetic study in stable renal transplant patients, all three innovator tacrolimus formulations of tacrolimus were compared [16]. Conversion from tacrolimus immediate release to Envarsus^®^ required a 30% reduction in the total daily dose, from tacrolimus immediate release to Advagraf^®^ an 8% increase in the total daily dose and from Advagraf^®^ to Envarsus^®^ a 36% decrease in the total daily tacrolimus dose. These percentages are averages for the population, and in some patients the required change in dose to maintain a stable blood level may be substantially lower or higher. An important implication is that the formulations are not interchangeable and that uncontrolled switching between these formulations can potentially lead to clinically relevant changes in tacrolimus exposure and serious patient harm [17]. In case of a switch from one formulation to another, intensified therapeutic drug monitoring is warranted [18].

### 1.2. Tacrolimus Trough Concentration Versus AUC, and the Influence of Genotype

In daily practice in the vast majority of patients on tacrolimus treatment, the dose is adjusted based on the monitoring of trough (predose) concentrations. Troughs are used because they are convenient for both the patient and the health care provider, and it is assumed that the correlation between the trough and area-under-the-concentration-versus-time-curve (AUC) is good. However, the correlation between the AUC and trough is variable, and sequentially monitoring trough concentrations may not always give a good indication of overall drug exposure. Although in general most experts would agree that therapeutic drug monitoring based on the AUC may lead to an improved outcome, there is no evidence for this assumption, as prospective randomized clinical trials comparing trough versus AUC monitoring have not been performed [19].

However, a lack of evidence does not mean that the assumption is false. By limiting monitoring to troughs only, high peak concentrations will go unnoticed. Especially for patients with a high dose, the peak concentrations may reach higher values, and the higher peaks may be related to tacrolimus-induced toxicity [20]. Patients with a *CYP3A5* expresser genotype are treated on average with higher doses. In these patients in particular, modified release formulations may avoid higher peaks [21]. For Envarsus^®^, it has been shown that, on average, the difference between trough and peak concentrations (referred to as peak-trough fluctuations) is smaller than for the immediate release formulation [22]. Whether this really results in a better clinical outcome remains unclear. Comparative studies have not convincingly shown a reduced incidence of side effects, perhaps with the exception of a reduced incidence of tremor in patients switched to the Envarsus^®^ formulation [23].

## 2. CYP3A5 Genotype and Changes in Exposure between Immediate Release and Modified Release

Most studies comparing the pharmacokinetics of the different tacrolimus formulations are studies in which stable patients on maintenance treatment with the immediate release formulation of tacrolimus were switched to one of the modified release formulations (Table 1). Sometimes, the primary goal of the study was to investigate the pharmacokinetics, and often in these studies the investigators collected sufficient samples to calculate or estimate the AUC. In studies where the primary outcome was a clinical parameter (changes in renal function, incidence of rejection, incidence of side effects), the pharmacokinetic data were often limited to trough concentrations only.

In a population pharmacokinetic analysis, Benkali et al. [14] studied the influence of several patient characteristics on the pharmacokinetics of modified release tacrolimus (Advagraf^®^). In a group of 41 patients, they found that the *CYP3A5* genotype was the only covariate retained in the final model, with a two-fold higher apparent clearance of tacrolimus in expressers (with the *CYP3A5*1/*1* and *CYP3A5*1/*3* genotypes) than in nonexpressers (with the *CYP3A5*3/*3* genotype). These data confirmed what was already known from the impact of the *CYP3A5* genotype on immediate release tacrolimus.

A German group reported that following a switch from the twice-daily immediate release formulation (Prograf^®^) to the once-daily modified release tacrolimus (Advagraf^®^), patients had, on average, a significantly lower tacrolimus trough concentration and dose-normalized trough concentration (14%, *p* = 0.0004 and 23%, *p* = 0.001, respectively) [24]. Although the number of patients in this largely Caucasian population of renal transplant recipients was small (41), they did find a significant influence of CYP3A5 expression on the change in tacrolimus exposure. The tacrolimus concentration remained almost constant in CYP3A5 expressers, whereas the trough concentration and dose-normalized trough concentration decreased significantly in nonexpressers (16%, *p* = 0.001 and 25%, *p* = 0.006).

In a French, prospective, single-center, open-label study on stable kidney transplant patients, 17 CYP3A5 expressers and 15 nonexpressers were switched from immediate release tacrolimus to modified release tacrolimus (Advagraf^®^) [25]. Not surprisingly, the investigators found that for both formulations the mean tacrolimus daily dose was significantly higher and the dose-adjusted AUC24 was significantly lower in the CYP3A5 expresser group. More remarkable was their observation that, following the switch to modified release tacrolimus, there was a significant decrease in the mean tacrolimus trough concentrations in the CYP3A5 expressers, while they remained stable in the nonexpressers. The effect of the genotype remained unexplained, and the authors suggested to even more carefully monitor CYP3A5 expressers after a switch.

In a study from Canada, following a switch from immediate release tacrolimus to modified release tacrolimus (Advagraf^®^) in stable kidney transplant patients, dose increases of more than 30%, necessary to maintain tacrolimus exposure, were most frequent in patients originating from East Asia [26]. Although the genotype was not assessed, this is suggestive of a pharmacogenetic effect (as about half of Asian patients have the CYP3A5 expresser). In a study in Japan, Niioka and Satoh et al. [27] confirmed that, in CYP3A5 expressers, there is a stronger decrease in the dose-adjusted tacrolimus AUC after switching to a once-daily formulation [27,28].

With the exception of the German study, the published data suggest that, in patients expressing CYP3A5, the change in tacrolimus exposure after switching from Prograf^®^ to Advagraf^®^ is larger than in those not expressing CYP3A5. Although it is tempting to conclude that this is due to the differential expression of CYP enzymes along the gut wall, it is fair to say that many pieces of the puzzle are still missing. The studies included in our review are all limited by a small sample size, and the duration of the follow-up after the switch is short. Furthermore, liver function (other than the genotype) was not formally assessed in the patients included in these studies.

In 2018, Trofe–Clark et al. [29] published a randomized prospective crossover study in African American kidney transplant patients on stable maintenance therapy with Prograf^®^ (or a twice-daily generic formulation). The study is also known as the ASERTAA-study [A Study of Extended Release Tacrolimus in African Americans]. Patients were randomly assigned to continue Prograf^®^ on days 1 to 7 and then switch to Envarsus^®^ on day 8 or receive Envarsus^®^ on days 1 to 7 and then switch back to Prograf^®^ on day 8. The once-daily Envarsus^®^ tablets were dosed 15% lower than the total daily Prograf^®^ dose. Full 24 h AUCs were collected on days 7, 14 and 21. A total of 46 patients completed the entire pharmacokinetics study of three 24-h assessments, and of these 35 (76%) were CYP3A5 expressers. The study showed that, during treatment with Prograf^®^, the higher tacrolimus dose in CYP3A5 expressers resulted in a high Cmax, whereas during treatment with Envarsus^®^ the shape of the pharmacokinetic profile was not affected by the *CYP3A5* genotype.

We need to improve our understanding of the full implications of pharmacogenetic variants affecting the expression and function of drug transporters in the gut wall and of enzymes involved in phase I and phase II metabolism on the different formulations [1]. For example, it is unclear why a switch from the tacrolimus immediate release to the modified release formulation Advagraf^®^ is associated with an 8% increase in the total daily dose, whereas a switch to Envarsus^®^ leads, on average, to a 24% decrease in the total daily tacrolimus dose. The decrease in CYP3A activity towards the more distal part of the small intestine and the colon may explain the higher bioavailability of Envarsus^®^ compared to Prograf^®^, but this does not explain why, for the switch to Advagraf^®^, a small dose increase is typically required to maintain tacrolimus concentrations at the same level. It is important to note that a comparison of pharmacokinetic profiles between the three formulations shows that Advagraf^®^ is characterized by only a slightly longer Tmax compared to Prograf^®^, while Envarsus^®^ has a significantly longer Tmax. The implication is that Envarsus^®^ may and Advagraf^®^ may not be largely absorbed in the more distal parts of the small intestine. Another potential explanation is that the higher bioavailability of Envarsus^®^ is not related to the distribution of CYP3A enzymes along the gut wall but is the result of the fact that MeltDose technology reduces the drug to the smallest possible particle size, allowing for a better dissolution and absorption. A better understanding of these processes will not only give more insights into the pharmacokinetic changes after switching from one tacrolimus formulation to another, but additionally, for other drugs, this will provide important information on the effects of novel formulation technologies on solubility, presystemic metabolism and bioavailability. Furthermore, in situations with increased intestinal motility, the residence time of drugs in the proximal parts of the small intestine may be much shorter, resulting in less metabolism in the intestinal wall and increased bioavailability.Finally, the susceptibility to drug-drug interactions may differ between the different tacrolimus formulations. It was already shown that CYP3A5 expressers are relatively resistant to the effect of the azole-induced inhibition of CYP3A activity [30].

**Table 1 genes-11-01205-t001:** Effect of CYP3A5 genotype on exposure to tacrolimus when switching from one formulation to another.

Author	Dosage Form	Study Design	Patients	Main Results
Benkali et al., 2010 [31]	Advagraf^®^	Population pharmacokinetic model	Renal transplant patients *CYP3A5*1/1* n = 1 *CYP3A5*1/3* n = 4 *CYP3A5*3/3* n = 36	The apparent clearance was two-fold higher in expressers (*CYP3A5*1/1* and *CYP3A5*1/3)* than in nonexpressers *(CYP3A5*3/3)* The *CYP3A5* genotype explained 25% of the interindividual variability in apparent clearance
Wehland et al., 2010 [23]	Switch from Prograf^®^ to Advagraf^®^	Prospective, single-center switch study	Renal transplant patients *CYP3A5*1/1* n = 0 *CYP3A5*1/3* n = 13 *CYP3A5*3/3* n = 27	After the conversion, mean tacrolimus trough levels and dose-normalized trough level remained almost constant in *CYP3A5*1/*3* patients, but decreased significantly in *CYP3A5*3/*3* patients (16%, *p* = 0.001 and 25%, *p* = 0.006)
Glowacki et al., 2011 [24]	Switch from Prograf^®^ to Advagraf^®^	Prospective, single-center, switch study	Renal transplant patients *CYP3A5*1/1* n = 2 *CYP3A5*1/3* n = 14 *CYP3A5*3/3* n = 15	In the nonexpressor group, mean blood trough concentration was comparable for both formulations while it decreased significantly in the expressor group after the switch (8.2 ± 2.2 vs. 6.3 ± 2.5 ng/mL, *p* = 0.002)
Glick et al., 2014 [25]	Switch from Prograf^®^ to Advagraf^®^	Prospective cohort study	Renal transplant patients Caucasian n = 282 East Asia n = 91 South Asia n = 75 African Canadian n = 18 Middle Eastern n = 12 Other n = 10	The percentage of patients requiring a dose increase of 30% or greater varied from 8.0% for South Asians to 27.5% for East Asians (*p* = 0.03)
Niioka et al., 2012 [26]	Advagraf ^®^ Prograf ^®^	Retrospective noncontrolled single-center study	Renal transplant patients Advagraf^®^ *CYP3A5*1/1* n = 4 *CYP3A5*1/3* n = 6 *CYP3A5*3/3* n = 15 Prograf ^®^ *CYP3A5*1/1* n = 7 *CYP3A5*1/3* n = 20 *CYP3A5*3/3* n = 20	Dose adjusted AUC was approximately 25% lower for Advagraf^®^ than Prograf^®^ in patients carrying the *CYP3A5*1* allele
Satoh et al., 2014 [27]	Advagraf ^®^ Prograf ^®^	Retrospective, single-center study	Advagraf^®^ *CYP3A5*1/1 + CYP3A5*1/3* n = 9 *CYP3A5*3/3* n = 15 Prograf ^®^ *CYP3A5*1/1* + *CYP3A5*1/3* n = 18 *CYP3A5*3/3* n = 14	Dose adjusted AUC was approximately 25% lower for Advagraf^®^ than Prograf^®^ in CYP3A5 expressers during the study period.
Trofe-Clark et al. 2018 [29]	Switch from Envarsus^®^ to Prograf^®^ or Switch from Prograf^®^ to Envarsus^®^	Randomized prospective crossover study	Renal transplant patients *CYP3A5* expressers *CYP3A5*1/1* n = 12 *CYP3A5*1/3* n = 17 *CYP3A5*1/6* n = 6 *CYP3A5* nonexpressers *CYP3A5*3/3* n = 4 *CYP3A5*3/6* n = 6 *CYP3A5*6/6* n = 1	Cmax was 33% higher for Prograf^®^ in *CYP3A5* expressers compared with nonexpressers (*p* = 0.04). With Envarsus^®^ the difference was 11% (*p* = 0.4).

Advagraf ^®^ = once-daily modified release tacrolimus. Envarsus ^®^ = once-daily modified release tacrolimus. Prograf ^®^ = twice-daily immediate release tacrolimus.

## 3. Conclusions

Immediate release and modified release formulations of tacrolimus are widely available. Most clinicians are aware of the fact that, in the case of switching from one formulation to another on a 1:1 mg basis, changes in tacrolimus exposure can be expected. Therefore, at the time of switching a dose adjustment should already be implemented, especially if the Envarsus^®^ formulation is involved.

What is not common knowledge is that the change in drug exposure following a switch depends on the genotype of the patient. Most studies suggest a bigger change in tacrolimus trough concentrations in CYP3A5 expressers (CYP3A5*1 allele carriers) compared to CYP3A5 nonexpressers (CYP3A5*3/*3). In these patients, intensified therapeutic drug monitoring involving checking drug levels within one week after switching, is recommended.

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
