# Peer review of "Comparison of the Impact of Pharmacogenetic Variability on the PK of Slow Release and Immediate Release Tacrolimus Formulations"

_genes, 2020, doi:10.3390/genes11101205_

Round 1
Reviewer 1 Report
In this review van Gelder et al. discussed about the role of CYP3A5 genotypes on the PK of tacrolimus different formulations (slow or immediate release). The analysis does not give an innovative contribution to the field compared to already reported in the literature. Furthermore, the conclusions of this review are not very clear indeed they seem contradictory. it is requested to better explain the results of the evaluation of the studies. In addition, authors should discuss about the role of CYP3A5 in the interaction of tacrolimus with other drugs such as fluconazole, which may be used in these patients and which might require dose variations. The authors must discuss about the limit of sample size of analyzed studies, the duration of follow-up and patient's liver function evaluations in these studies which would require larger scale studies. Discuss about the role of multiple genetic polymorphisms on prediction of tacrolimus metabolism beyond CYP3A5 genotype.
Author Response
Reply to reviewer #1
The scope of our paper is specifically on the effect of CYP3A5 genotype on changes in exposure to tacrolimus when patients are switched from one formulation to another (once daily vs twice daily). We agree with the reviewer that there are a lot of published data on changes in tacrolimus concentrations when patients are switched from the (twice daily) Prograf formulation to the (once daily) Advagraf or (once daily) Envarsus formulation. However, rather than reviewing this literature, the novelty of our paper is that we review the impact of CYP3A5 genotype on the changes in tacrolimus concentrations, during such conversions. Although not all published studies reported the same outcome, the current data suggest that the changes in drug exposure are larger in patients who express the CYP3A5 enzyme (CYP3A5 *1/*3 or *1/*1) compared to non-expressers (CYP3A5*3/*3). This finding is not widely recognized and from personal experiences and contacts with other doctors we are quite sure that transplant physicians do not take this effect of genotype into account when they switch patients from one formulation to another. Therefore we are convinced our paper contributes to the current literature. To emphasize the objective of our review we have added the following line to the second paragraph of the introduction “The objective of this review is to discuss the effect of CYP3A5 genotype on changes in exposure to tacrolimus when patients are switched from one formulation to another (once daily vs twice daily)”
Other genotypes than CYP3A5 were not included as CYP3A5 genotype is by far the most relevant genotype in the metabolism of tacrolimus (see for example the broad ADME genotyping strategy with DMET Plus by Birdwell et al, published in Pharmacogenetics Genomics in 2012). Moreover to the best of our knowledge studies investigating the impact of polymorphisms in other genes on tacrolimus pharmacokinetics following a conversion from one formulation to another are not available.
Reviewer #1 interprets the conclusions as not very clear and seemingly contradictory. We agree that thus far it remains unexplained why patients may require a small daily dosage increase if converted from tacrolimus immediate release to Advagraf®, while a daily dosage reduction appears necessary for conversion from tacrolimus immediate release to Envarsus®. The decrease in CYP3A activity towards the more distal part of the small intestine and the colon may explain the higher bioavailability of Envarsus® compared to Prograf®, but this does not explain why for the conversion to Advagraf® a small dose increase is typically required to maintain tacrolimus concentrations at the same level. It is important to note that a comparison of pharmacokinetic profiles between the three formulations shows that Advagraf® is characterized by only a slightly longer Tmax compared to Prograf®, while Envarsus® has a significantly longer Tmax. The implication is that Envarsus® may, and Advagraf® may not be largely absorbed in the more distal parts of the small intestine. Another potential explanation is that the higher bioavailability of Envarsus® is not related to the distribution of CYP3A enzymes along the gut wall, but the result of the fact that MeltDose technology reduces the drug to the smallest possible particle size, allowing for better dissolution and absorption. This has now been added to the discussion part of the manuscript and hopefully provides a more consistent interpretation.
The suggestion that the susceptibility to drug-drug interactions may also differ between the different tacrolimus formulations has now been added to the last paragraph of the manuscript. We have also included a statement on the limitations of the studies reported in this review: “ The studies included in our review are limited by a small sample sample size, and the duration of follow-up after the switch is short. Also liver function (other than genotype) was not formally assessed in the patients included in these studies.”

Reviewer 2 Report
Good job on explaining the need to evaluate variables associated while changing formulations. That being said, the review severely lacks comprehensive referencing.
- Please clarifiy on addition in this review compared to previously published similiar kind of reviews such as, https://rdcu.be/b7ONJ (2017). Also, reference such previously piublised studies to provide the audience with a more detailed view.
- Also, please include more recent clinical studies related to tricolumous such as doi: 10.1007/s12325-019-00904-x (2019) and https://doi.org/10.1016/j.transproceed.2019.04.028 (2019)
- in the table include any study if available for the switch to/from Envarsus formulation
Author Response
Reply to reviewer #2
We thank the reviewer for his positive comments and suggestions to include additional papers.
Point 1. The review paper of Tremblay and Alloway (published in 2017) has now been added to the references, and in the introduction this paper is mentioned as a recommended review on the clinical development of tacrolimus, and the potential advantages of the subsequently developed alternative formulations. From the statement that “the review severely lacks comprehensive referencing” it seems that perhaps reviewer #2 has another perspective on the scope of this review. It is not our intention to write a review on all the studies that have compared clinical outcomes between patients on immediate release versus slow release tacrolimus formulations. To address this point we have added the following line to the second paragraph of the introduction “The objective of this review is to discuss the effect of CYP3A5 genotype on changes in exposure to tacrolimus when patients are switched from one formulation to another (once daily vs twice daily”
Point 2. The paper doi: 10.1007/s12325-019-00904-x (2019) does not really fit within the scope of this review. This paper reports a retrospective review of the clinical outcome data of 19 patients treated with the Advagraf® formulation, compared with 55 patients treated with Prograf®. There is no information in this paper on CYP3A5 genotype, or on changes in tacrolimus exposure when patients were switched from one formulation to the other. Therefore I have not included this paper.
The other paper has now been added, and is used in paragraph 2.1 as a reference for the statement that “Advagraf® is also not bio-equivalent to tacrolimus immediate release”.
Point 3. We have now added the paper by Trofe-Clark, as it reported a controlled conversion study comparing the pharmacokinetics of immediate release tacrolimus and Envarsus® in CYP3A5 expressers and non-expressers.

Round 2
Reviewer 1 Report
The authors responded satisfactorily to my comments.